# Loss of COX4I1 Leads to Combined Respiratory Chain Deficiency and Impaired Mitochondrial Protein Synthesis

**DOI:** 10.3390/cells10020369

**Published:** 2021-02-10

**Authors:** Kristýna Čunátová, David Pajuelo Reguera, Marek Vrbacký, Erika Fernández-Vizarra, Shujing Ding, Ian M. Fearnley, Massimo Zeviani, Josef Houštěk, Tomáš Mráček, Petr Pecina

**Affiliations:** 1Laboratory of Bioenergetics, Institute of Physiology, Czech Academy of Sciences, 142 00 Prague, Czech Republic; kristyna.cunatova@fgu.cas.cz (K.Č.); d.pajuel@gmail.com (D.P.R.); marek.vrbacky@fgu.cas.cz (M.V.); josef.houstek@fgu.cas.cz (J.H.); 2Department of Cell Biology, Faculty of Science, Charles University, 128 00 Prague, Czech Republic; 3MRC Mitochondrial Biology Unit, University of Cambridge, Cambridge, CB2 0XY, UK; Erika.Fernandez-Vizarra@glasgow.ac.uk (E.F.-V.); shujing.ding@mrc-mbu.cam.ac.uk (S.D.); ian.fearnley@mrc-mbu.cam.ac.uk (I.M.F.); massimo.zeviani@unipd.it (M.Z.)

**Keywords:** mitochondria, OXPHOS, cI, COX, cIV, COX4, knock-out, cIV assembly, complex I, biogenesis interdependency, complexome profiling, mitochondrial protein synthesis

## Abstract

The oxidative phosphorylation (OXPHOS) system localized in the inner mitochondrial membrane secures production of the majority of ATP in mammalian organisms. Individual OXPHOS complexes form supramolecular assemblies termed supercomplexes. The complexes are linked not only by their function but also by interdependency of individual complex biogenesis or maintenance. For instance, cytochrome *c* oxidase (cIV) or cytochrome *bc1* complex (cIII) deficiencies affect the level of fully assembled NADH dehydrogenase (cI) in monomeric as well as supercomplex forms. It was hypothesized that cI is affected at the level of enzyme assembly as well as at the level of cI stability and maintenance. However, the true nature of interdependency between cI and cIV is not fully understood yet. We used a HEK293 cellular model where the COX4 subunit was completely knocked out, serving as an ideal system to study interdependency of cI and cIV, as early phases of cIV assembly process were disrupted. Total absence of cIV was accompanied by profound deficiency of cI, documented by decrease in the levels of cI subunits and significantly reduced amount of assembled cI. Supercomplexes assembled from cI, cIII, and cIV were missing in COX4I1 knock-out (KO) due to loss of cIV and decrease in cI amount. Pulse-chase metabolic labeling of mitochondrial DNA (mtDNA)-encoded proteins uncovered a decrease in the translation of cIV and cI subunits. Moreover, partial impairment of mitochondrial protein synthesis correlated with decreased content of mitochondrial ribosomal proteins. In addition, complexome profiling revealed accumulation of cI assembly intermediates, indicating that cI biogenesis, rather than stability, was affected. We propose that attenuation of mitochondrial protein synthesis caused by cIV deficiency represents one of the mechanisms, which may impair biogenesis of cI.

## 1. Introduction

Energetic needs of mammalian organisms fully depend on ATP produced by oxidative phosphorylation (OXPHOS), a major metabolic pathway localized in the inner mitochondrial membrane. OXPHOS complexes are encoded by both mitochondrial (mtDNA) and nuclear (nDNA) DNA. Complex I (NADH dehydrogenase, cI) contains 7 out of 45 subunits encoded by mtDNA, complex II (succinate dehydrogenase, cII) contains 4 nuclear-encoded subunits, complex III (cytochrome *bc1* complex, cIII) contains 1 out of 10 subunits encoded by mtDNA, and complex IV (cytochrome *c* oxidase, cIV) contains 3 out of 14 subunits encoded by mtDNA. Regulation of OXPHOS capacity may therefore occur at various levels of mtDNA expression, including mRNA maturation, mRNA stability, translational coordination, ribosomal biogenesis, and translation itself [1,2,3,4,5]. Expression and posttranslational modifications of individual subunits in cytosol and mitochondria need to be coordinated by signaling crosstalk between nucleus and mitochondria [6]. Moreover, for proper OXPHOS function, many enzyme cofactors are needed and their homeostasis and efficient incorporation into apoenzyme are important [7]. Individual subunits and cofactors must assemble to produce the mature and functional enzymes, with a myriad of so-called “assembly factors” involved. In addition, respiratory chain complexes form supramolecular assemblies termed as supercomplexes, adding yet another level of complexity in OXPHOS biogenesis. The most abundant supercomplex in mammalian mitochondria was named as respirasome, because it is composed by all the elements capable of transferring electrons from NADH to oxygen, i.e., one copy of complex I, dimer of complex III, and one copy of cIV (I III_2_IV) [8]. Its structure has been recently solved with overall resolution at 5.8 Å [9] or 5.4 Å [10].

Mammalian cIV consists of a catalytic core composed of three mtDNA-encoded subunits: MT-CO1, MT-CO2, and MT-CO3, surrounded by 11 different regulatory subunits encoded in nucleus: COX4, COX5A, COX5B, COX6A, COX6B, COX6C, COX7A, COX7B, COX7C, COX8, and NDUFA4, which is currently also considered as part of the cIV, discovered on the basis of genetic [11] and structural data [12]. Six of the nuclear-encoded subunits exist as developmental stage or tissue-specific isoforms [13]. In case of COX4 subunit specifically, alternative isoforms COX4I1 and COX4I2 are expressed by independent genes. COX4I1 represents the ubiquitous isoform, while COX4I2 is mainly lung-specific and its expression is further regulated by oxygen levels [13]. It was recently shown that COX4I2 is necessary for hypoxic pulmonary vasoconstriction [14], and that COX4 isoform exchange may modulate cIV affinity to oxygen [15].

Biogenesis of cIV is a complicated and highly regulated process that was at first described as sequential incorporation of cIV subunits through four distinct assembly intermediates S1–S4, reflecting rate-limiting steps in enzyme biogenesis [16]. The cIV assembly mechanism is not spontaneous, because numerous accessory nuclear DNA-encoded proteins are needed to build the holoenzyme. Their function is required for all steps of the process and most of them are evolutionary conserved from lower to higher eukaryotes [17,18]. Currently, cIV assembly is considered to be modular, with independent maturation of three mtDNA-encoded subunits accompanied by addition of specific sets of nuclear-encoded subunits [4,19]. MT-CO1 maturation occurs within the MITRAC complexes (mitochondrial translation regulation assembly intermediates of cytochrome *c* oxidase) before binding of an assembly initiator subcomplex containing nuclear-encoded subunits COX4 and COX5A, the first ones to be assembled together [20]. The MT-CO2 module then associates with nascent cIV upon MT-CO2 copper metallization by the specific SCO1/SCO2, COX16, and COA6 chaperones [21,22,23,24,25]. Addition of the MT-CO3 module, with the help of HIGD2A, finalizes cIV assembly [4,26,27].

Complex I is the largest OXPHOS complex, which in mammals consists of 45 subunits, including 14 core subunits homologous to prokaryotic cI components, and numerous accessory subunits that are indispensable for enzyme assembly and stabilization [5]. The biogenesis of the complex is characterized by pre-assembly of individual modules that reflect enzyme functional units, aided by specific assembly factors [4]. The membrane arm containing the mtDNA-encoded subunits ND1-ND6 first associates with the ubiquinone (CoQ)-binding module. The assembly is completed by attachment of the soluble N-module responsible for electron transfer between NADH and CoQ.

It has been repeatedly found that deficiency in one of the complexes may in turn affect the other ones. Therefore, complexes are linked not only by their function, but also by interdependency in individual complex formation or maintenance. Notably, deficiency of cIII has been frequently shown to cause decrease of cI content and activity. This occurrence was interpreted as a consequence of cI destabilization in the absence of its supercomplex partner [28,29,30,31,32,33]. However, some studies convincingly show that the interdependency is associated with impairment in cI maturation, which can be at least partially attributed to essential association of both these complexes within the respirasome precursor during their assembly [34,35]. Although less frequently, it has been shown that cIV deficiency also affects the level of assembled cI, both non-associated and within supercomplexes. Interestingly, this phenomenon seems to occur only in rare cases with complete absence of cIV caused by defect in early phases of MT-CO1 maturation/assembly [36,37]. These studies indicate that cI is affected at the level of complex assembly, suggesting some degree of interdependency between biogenesis of these respiratory chain complexes. In contrast, another reported case of cI-cIV interdependency postulated that cIV loss due to nonsense mutations in MT-CO1 gene led to increased activity of the mitochondrial m-AAA protease AFG3L2, which in turn impaired the stability of complex I [38]. Importantly, numerous cases of milder cIV assembly defects with at least minimal residual portion of assembled cIV display no effect on cI [39]. Recently, a study using 143B ΔMT-CO1 and ΔMT-CO2 cybrid cells demonstrated that even an incomplete, non-canonical MT-CO1 submodule containing subunits COX5B and HIGD2A can associate with the I III_2_ supercomplex and stabilize it, thus maintaining the content of these respiratory complexes [40].

Here, we explored the mechanisms of cIV-cI interdependency by interfering with the earliest phases of cIV assembly using COX4 loss-of-function cellular models. We report that removal of COX4 subunit in HEK293 results in complete cIV deficiency. Furthermore, the impairment of the earliest steps in cIV assembly also causes profound deficiency of complex I, which is associated with decreased synthesis of mtDNA-encoded cI subunits as well as reduced content of mitochondrial ribosomal proteins. We propose that mitochondrial protein synthesis attenuation may represent yet another mechanism of cIV-cI biogenesis interdependency.

## 2. Materials and Methods

### 2.1. Cell Culture and Models

Preparation of HEK293 (ATCC CRL-1573)-based cellular models, i.e., COX4I1, COX4I2, and COX4I1/4I2 gene knock-outs and cells expressing COX4I1 or COX4I2 on the background of COX4I1/4I2 knock-out (KO) HEK293 was described in detail in our previous study [15]. Briefly, COX4I1 and COX4I2 knock-outs were introduced by CRISPR (Clustered Regularly Interspaced Short Palindromic Repeats) knock-out technology employing Cas9-D10A paired nickase with two chimeric RNA duplexes [41]. To obtain double knock-out of both isoforms (COX4I1/4I2 KO), we subsequently knocked out COX4I2 and COX4I1. For this study, two individual clones of each knock-out was used (COX4I1 KO 1 and 2, COX4I2 KO 1 and 2, COX4I1/4I2 KO 1 and 2). For rescue experiments, COX4I1/4I2 KO 2 served as a background generating cell lines with stable expression of COX4I1 (COX4I1 KI) or COX4I2 (COX4I2 KI), using full length complementary DNA (cDNA) constructs in pcDNA3.1^+^ with C-terminal FLAG (DYKDDDDK) tag.

Cells were cultivated under standard conditions (37 °C, 5% CO_2_ atmosphere in Dulbecco’s Modified Eagle Medium: Nutrient Mixture F-12 (DMEM/F-12) (Thermo Fisher Scientific, 31331-028), smented with 10% (*v*/*v*) fetal bovine serum (FBS) (Thermo Fisher Scientific, 10270-106), 40 mM 4-(2-hydroxyethyl)-1-piperazineethanesulfonic acid (HEPES), 50 µM uridine, and antibiotics (100 U/mL penicillin + 100 μg/mL streptomycin (Thermo Fisher Scientific, 15140-122)).

### 2.2. SDS-PAGE

Tricine-sodium dodecyl sulphate polyacrylamide gel electrophoresis (SDS-PAGE) was used for separation of proteins on the basis of their different molecular weights under denaturing conditions. SDS-PAGE samples were prepared from frozen aliquots of harvested cellular pellets according to [42], to a final concentration 2–5 mg protein/mL. Then, samples were sonicated and incubated for 20 min at 40 °C. Samples (10–50 µg of protein as indicated) were separated on 12% gel using Mini-PROTEAN III apparatus (Bio-Rad, Hercules, CA, USA).

### 2.3. Native Electrophoresis

Blue-native (BN)-PAGE electrophoresis [43] was used to separate native protein complexes. For protein complex solubilization, mild non-ionic detergent digitonin, which prevents dissociation of supercomplexes, was used; thus, OXPHOS complexes and their supercomplexes could be detected. Mitochondria from HEK293 were released by hypotonic shock and isolated using differential centrifugation as previously described [44] and then stored as dry pellets at -80 °C. Isolated mitochondrial pellets were solubilized by digitonin (6 g detergent/g of protein) and prepared for analysis according to [43]. Samples were separated on 5–16% polyacrylamide gradient gel using Mini-PROTEAN III apparatus (Bio-Rad).

### 2.4. Western Blot (WB) and Immunodetection

Proteins separated by electrophoresis were transferred onto PVDF (polyvinylidene difluoride) membrane (Immobilon FL 0.45 µm, Merck) by semi-dry electroblotting (0.8 mA/cm^2^, 1 h) using Transblot SD apparatus (Bio-Rad). Immunodetection was performed as described in [15]. Primary and secondary antibodies used for immunodetection are listed in Appendix A. Signal detection was performed using fluorescence scanner Odyssey (LI-COR Biosciences, Lincoln, NE, USA). Signals detected by scanner were quantified by software AIDA Image Analyzer (Raytest Isotopenmessgeräte GmbH, Germany).

### 2.5. Mass Spectrometry—Label-Free Quantification (MS LFQ)

Biological duplicates (COX4I1 KO clones 1 and 2, COX4I2 KO clones 1 and 2, COX4I1/4I2 KO clones 1 and 2) were harvested and MS LFQ analysis was performed in technical duplicates, as described previously [45]. Briefly, cell pellets (100 µg of protein) of wild-type HEK293 cell line, COX4I1, COX4I2, and COX4I1/4I2 KO clones were solubilized by sodium deoxycholate (final concentration 1% (*w*/*v*)), reduced with TCEP ([tris(2-carboxyethyl)phosphine]), alkylated with MMTS (S-methyl methanethiosulfonate), digested sequentially with Lys-C and trypsin, and extracted with ethylacetate saturated with water, as previously described. Samples were desalted on Empore C18 columns, dried in Speedvac, and dissolved in 0.1% trifluoroacetic acid (TFA) + 2% acetonitrile. About 1 µg of peptide digests were separated on a 50 cm C18 column using 2.5 h elution gradient and analyzed in a data dependent acquisition (DDA) mode on an Orbitrap Fusion Tribrid (Thermo Fisher Scientific) mass spectrometer. Resulting raw files were processed in MaxQuant (v. 1.5.3.28) with label-free quantification (LFQ) algorithm MaxLFQ. Downstream analysis and visualization were performed in Perseus (maxquant.org/perseus/, v. 1.5.8.5). LFQ dataset is publicly available in the PRIDE repository (https://www.ebi.ac.uk/pride), accession number PXD023367 (accessed on 15 February 2021).

### 2.6. Stable Isotope Labeling Using Amino Acids in Cell Culture (SILAC)/Complexome Profiling

Wild-type and COX4I1/4I2 KO clone 2 cells were cultivated in “heavy” (^15^N- and ^13^C-labeled Arg and Lys) and in “light” (^14^N and ^12^C Arg and Lys) DMEM (Thermo Fisher Scientific). Unlabeled (“light”) cell line was mixed equally with labeled (“heavy”) one and mitochondria-enriched fraction was isolated [46]. Mitochondrial pellets were solubilized by digitonin and Bis/Tris BN-PAGE was performed using 3–12% polyacrylamide gradient gel. The gel was stained, fixed, and excised in 64 slices. Slices were in-gel digested by trypsin and analyzed by tandem mass spectrometry [19,47]. For the protein identification, peptide quantification and plain text export, Proteome Discoverer software (Thermo Fisher Scientific), and Mascot peptide search engine (Matrix Science) were used. For further data evaluation, Python code was used to separate unlabeled and labeled peptides into two datasets and R code was used to determine the relative peptide intensity for each protein according to the most frequently occurring representative peptide in both unlabeled and labeled forms (for details see [29]). The relative peptide intensity profile of each protein was normalized to the slice with the highest intensity from the average of duplicate experiments and was visualized in heatmaps using Microsoft Excel and Prism 8 (GraphPad Software, La Jolla California USA). Complexome profiling dataset is publicly available in the CEDAR database [48] (https://www3.cmbi.umcn.nl/cedar/browse), accession number CRX25 (accessed on 22 December 2020).

### 2.7. Mitochondrial Protein Synthesis Pulse-Chase Analysis

To study synthesis and turnover of mtDNA-encoded proteins, we essentially employed the method of in vivo metabolic pulse-chase protein labeling by ^35^S-methionine + ^35^S-cysteine as described [49], using Express [^35^S] Protein Labeling Mix (PerkinElmer, USA). For this experiment, wild-type HEK293 and two independent clones of COX4I1 KO, COX4I2 KO, and COX4I1/4I2 KO were analyzed in two independent experiments. Cells were labeled in DMEM, without methionine and cysteine, in the presence of 100 μg/mL of anisomycin and the radioactive labeling mix with 100 μCi/mL for 3 hours. Afterwards, 20 μL of the mixture of “cold” methionine and cysteine (non-labeled Met and Cys) was added into each well (final concentration 250 μM) and incubated for a further 15 min. Then, cells were washed two times with phosphate buffered saline PBS containing 250 μM cold methionine and 250 μM cold cysteine. One aliquot of labeled cells was immediately harvested (“pulse” sample). Parallel dishes of labeled cells (“chase” samples) were further incubated in complete non-labeling DMEM medium for 24 hours. Cellular pellets were later used for SDS-PAGE analysis using large format Hoefer SE 600 Chroma Vertical Electrophoresis System (Thermo Fisher Scientific). The 70 µg protein aliquots of each sample were separated by 16% gel. Following the electrophoretic run, gel was fixed in staining solution (40% (*v*/*v*) methanol, 8% (*v*/*v*) acetic acid, 0.05% (*w*/*w*) Coommasie Brilliant Blue R-250) and dried onto Whatmann CHR 3 mm paper. Radioactivity was detected by exposing the gel to Storage Phosphor Screen BAS-IP SR 2025 E for 14 days (GE Healthcare), which was then scanned by FX Molecular Imager (Bio-Rad). Signals detected by scanner were quantified by software AIDA Image Analyzer.

### 2.8. Measurement of Metabolic Fluxes

Parallel measurement of mitochondrial respiration and glycolytic rate was performed using Seahorse Extracellular Flux (XF) Analyzer (Agilent Technologies, USA), as described previously [15]. We analyzed wild-type HEK293 (*n* = 5 biological replicates, measurement on different days), COX4I1 clone 1 (*n* = 5), COX4I2 KO clone 1 (*n* = 4), and COX4I1/COX4I2 clone 1 (*n* = 3). Briefly, 1 day prior to measurement, 3.10^4^ cells were seeded in pentaplicates in poly-l-lysine-coated wells of measuring plate, and incubated overnight under standard cultivation conditions. For the measurement, the cells were washed with 1 mL of XF Assay Medium Modified DMEM containing 10 mM glucose as a substrate (pH 7.4, 37 °C), and 500 μL of the same medium with 0.2% (*w*/*w*) BSA was added; then, the microplate was incubated at 37 °C for 30 min. Meanwhile, XFe24 Sensor Cartridge was prepared by injection of substrates and inhibitors to record basal metabolic rate with glucose and after subsequent additions reaching final concentrations of 1 µM oligomycin (Oligo); 1 µM carbonyl cyanide-4-(trifluoromethoxy)phenylhydrazone (FCCP); and inhibitor cocktail of 1 µM rotenone, 1 µg/mL antimycin A, and 100 mM 2-deoxyglucose (Rot+AA+2DG). For precise normalization of rates according to cell counts, immediately after Seahorse measurements, we stained cell nuclei by Hoechst dye (final concentration 5 µg/mL, diluted in FluoroBrite DMEM Media). Images of whole wells were acquired by Cytation 3 Cell Imaging Reader (BioTek, Winooski, VT, USA) and analyzed using software Gen5 (BioTek) to obtain cell counts for each well.

## 3. Results

### 3.1. Decreased Steady-State Content of cIV and cI Subunits in Cells Lacking COX4I1

We decided to elucidate how the complete knockout of the COX4 subunit interferes with biogenesis of cIV as well as other OXPHOS complexes. To study this, we utilized COX4I1 and COX4I2 genetic knock-outs in HEK293, and two clones of each mutant (COX4I1, COX4I2, and COX4I1/4I2 with loss of both isoforms) were used in subsequent experiments.

The steady-state content of five representative cIV subunits (COX4I1, MT-CO1, MT-CO2, COX5A, and COX6C) was examined by denaturing SDS-PAGE followed by Western blot analysis and immunodetection in two representative clones of each COX4I2 KO, COX4I1 KO, and COX4I1/4I2 KO in comparison with wild-type (wt) HEK293. Representative WB images are shown in Figure 1a, where COX4I1 and MT-CO2 were undetectable, COX5A and COX6C only barely detectable, and MT-CO1 significantly decreased in COX4I1 KO and double COX4I1/4I2 KO (Figure 1a,b, Appendix A). In contrast, COX4I2 KO cells showed a similar pattern to the wt. Densitometric quantification revealed that relative levels of studied cIV subunits in COX4I2 KO clones ranged between 55 and 120% of controls with significant decrease of MT-CO2 content when data were normalized to actin levels (Appendix A). However, upon normalization to citrate synthase levels (CS, marker of mitochondrial mass), cIV subunits content was unaffected (Figure 1b). Importantly, COX4I1 KO in HEK293 cells did not trigger expression of the alternative isoform COX4I2 (Figure 1a). The functionality and specificity of the antibodies used to detect COX4I1 or COX4I2 was confirmed using positive control samples from COX4I1/4I2 KO transfected with the cDNA of the corresponding genes—knock-in (KI) of either COX4I1 or COX4I2 isoforms (Figure 1c). As we showed previously [15], HEK293 cells (wt) do not show detectable levels COX4I2 protein. On the contrary, COX4I1 isoform was detected by both monoclonal (COX4I1) and polyclonal (dual COX4 isoform-specific) antibodies. COX4I2 protein was detected only in the positive control COX4I2 knock-in cell line (Figure 1c).

In addition to classical electrophoretic and immunodetection approach, relative levels of cIV subunits were assessed in a high-throughput manner using mass spectrometry label-free quantification (MS LFQ). Analysis of two technical replicates of two representative clones for each COX4I2 KO, COX4I1 KO, and COX4I1/4I2 KO yielded reliable data for nine cIV subunits (Figure 1d). These confirmed our previous findings of generalized cIV subunit deficiency in COX4I1 KO and COX4I1/4I2 KO cells, with the most affected subunit besides the KO-targeted COX4I1 being MT-CO2. COX4I2 KO clones were not significantly changed as cIV subunit amounts were comparable to wild-type cells (Figure 1d).

Steady-state levels of representative subunits of other OXPHOS complexes (NDUFA9 of cI, SDHA of cII, UQCRC2 (CORE2) of cIII, and ATP5F1A (F_1_α) and ATP5MC1 (F_o_c) of cV) were also examined by SDS-PAGE with Western blot immunodetection (Figure 1e). Differences between model cell lines were observed in protein level of NDUFA9, which was decreased in COX4I1, COX4I1/4I2, and COX4I2 KO clones when data were normalized to actin (Appendix A). Upon normalization to citrate synthase, NDUFA9 was at a significant level approximately twofold lower, but only in the COX4I1/4I2 KO cells (Figure 1f). UQCRC2 subunit was also decreased in COX4I1/4I2 KO, but not in COX4I1 single KO. On the other hand, cII and cV subunits’ levels were similar or even moderately increased in COX4I1/4I2 and COX4I1 KO cells (Figure 1e, Appendix A).

The Western blot-based quantification data were complemented by MS LFQ analysis of the complete cellular proteome. Overall, we observed only slightly decreased quantity of MitoCarta-annotated proteins in COX4I1/4I2 KO (Appendix A) and COX4I1 KO (Appendix A), which indicates that in spite of the cIV impairment, proteins can still be efficiently imported into mitochondria. However, it is also obvious that levels of cI subunits show profound and uniform decrease in COX4I1/4I2 and COX4I1 KO (Appendix A). Data for individual subunits of OXPHOS complexes are presented as heatmaps (Figure 1g), quantitating 38 cI subunits, all 4 subunits of cII, 8 subunits of cIII, and 12 subunits of cV. From the perspective of multi-subunit comparison, the decrease of NDUFA9 shown by Western blot analysis was confirmed by MS LFQ analysis. The levels of cI subunits were between 1.5- and 3-fold decreased in COX4I1 and COX4I1/4I2 KO clones, while levels of cI subunits in COX4I2 KO were comparable to wild-type. On average, levels of cIII subunits were slightly decreased in COX4I1/4I2 KO cells, but it was less apparent in COX4I1 KO (Figure 1g, Appendix A). The remaining OXPHOS complexes, cII and cV, were not altered in the COX4I1 and COX4I1/4I2 KO clones. Consistently, the levels of mitochondrial proteins overall, as well as of individual OXPHOS complexes in COX4I2 KO clones were not significantly affected, with the exception of cIII that was slightly decreased (Figure 1g, Appendix A).

### 3.2. Compromised OXPHOS Function in COX4I1-Lacking Cells Was Compensated by Increased Glycolysis

To characterize the bioenergetic phenotype of the KO clones, we examined the rates of mitochondrial respiration and glycolysis (Figure 2). After recording basal mitochondrial respiration (oxygen consumption rate (OCR); Figure 2a) and glycolysis (extracellular acidification rate (ECAR); Figure 2b), addition of oligomycin (inhibitor of cV) caused decrease of OCR in wild-type HEK293 as oxygen consumption coupled to ATP synthesis was inhibited and compensated by increased glycolytic activity represented by ECAR. Addition of chemical protonophore FCCP (which uncouples ADP phosphorylation and oxidation processes) increased OCR, allowing us to determine maximal electron transport chain (ETC) capacity. After final addition of inhibitors rotenone, antimycin A, and 2-deoxyglucose, both OXPHOS and glycolysis were inhibited, and therefore OCR and ECAR decreased to minimum and residual background rates were recorded.

COX4I2 KO pattern of respiratory and glycolytic fluxes was comparable to wild-type (Figure 2a,b), the OCR was even slightly increased compared to wild-type (Figure 2c), while ECAR was not significantly changed in COX4I2 KO (Figure 2d). Consistent with the profound decrease in cIV subunit levels, COX4I1 and COX4I1/4I2 KO clones showed complete absence of OXPHOS activity (OCR) or response to additions of uncoupler and inhibitors (Figure 2a). Thus, ATP production was independent of OXPHOS. Complete impairment of OXPHOS function was compensated by significantly increased glycolytic activity (ECAR) compared to wt cells (Figure 2b,d). Moreover, ECAR displayed no response to oligomycin, indicating that energy requirements of cells lacking COX4I1 are met by high steady-state glycolytic rates, likely operating near its full capacity. ECAR seemed to be higher for COX4I1/4I2 KO clone than for COX4I1 KO clone, but this probably reflected unique metabolic settings of individual KO clones.

Importantly, the respiratory OXPHOS impairment can be complemented by knock-in (KI) of either COX4I1 or COX4I2 isoforms of cIV subunit 4. Expression of either isoform on the COX4I1/4I2 KO background fully restored the respiratory pattern to match control cells, as was already shown by respirometric data in our previous study [25].

### 3.3. Effect of COX4 KO on OXPHOS Complexes Content and Assembly Status

To examine effect of OXPHOS subunits changes caused by COX4 isoforms knock-outs, we studied content and assembly status of OXPHOS complexes in native state by BN-PAGE followed by Western blot and immunodetection (Figure 3).

Immunodetection with an antibody against MT-CO1 subunit revealed the total absence of all assembled forms of cIV in COX4I1 and COX4I1/4I2 KO clones. Residual MT-CO1 subunit was detected with apparent migration around 130 kDa, likely representing the S1 (early MITRAC) assembly intermediate (Figure 3a). Early incorporating subunit COX5A, which also showed residual presence in SDS/PAGE experiments, was not detected in association with MT-CO1 in COX4I1 or COX4I1/COX4I2 KO clones (not shown). In contrast, COX4I2 KO displayed a band pattern and amount of cIV-containing complexes comparable to wild-type, including cIV monomer (IV), dimer (IV_2_), and III_2_IV and I III_2_IV supercomplexes (Figure 3a). cI detection under native conditions using NDUFA9 antibody also confirmed the findings from SDS/PAGE and MS LFQ, showing a profound decrease in fully assembled cI in COX4I1 and COX4I1/4I2 KO clones. Moreover, while cI was detected mainly in supercomplexes (I III_2_IV) in wild-type and COX4I2 KO, the remaining cI in COX4I1-lacking clones was mainly present in association with cIII (I III_2_) or detected as a faint band corresponding to a cI assembly intermediate (sub I) (Figure 3b). Detection of cIII-containing complexes using an antibody against UQCRC2 subunit showed major changes in complex distribution within various assembly forms as a result of COX4I1 absence, rather than changes in total enzyme content. In wild-type and COX4I2 KO cells, cIII was detected as dimer (III_2_), the III_2_IV complex, and in supercomplexes. In contrast, lack of cIV following COX4I1 gene knock-out shifted almost all of cIII to non-associated dimer (III_2_), and minor portion was bound to the residual cI in (I III_2_) (Figure 3c). A mild effect of cIV absence was also observed on cV, detected by ATP5F1B antibody, which showed a lower amount of cV dimer, and also minor accumulation of F_1_ and F_1_-c subassemblies in COX4I1 and COX4I1/4I2 KO clones (Figure 3d).

WB analysis of BN-PAGE revealed that content of complexes I and IV, as well as their ability to form supercomplexes was restored in cells expressing either COX4 isoforms on the background of COX4I1/4I2 KO (Figure 3e). Comparable patterns of the bands detected by both MT-CO1 and specific COX4I2 antibody in the COX4I2 KI cells confirmed the efficiency of isoform 2 to act in de novo complex assembly and its incorporation into supercomplexes (Appendix A).

The content and assembly status of respiratory chain complexes was further examined in COX4I1/4I2 KO (clone 2) by SILAC/complexome profiling analysis, comparing the amounts and assembly status to those of the control HEK293 (wt) (Figure 4). Due to the profound cIV deficiency in these cells, only two cIV subunits, COX5B and NDUFA4, were detected in the KO sample. Both subunits migrated in the low molecular weight region, thus representing unassembled subunits or early assembly intermediates. cIV-associated proteins HIGD1A and HIGD2A retained robust signal in KO cells, where a portion of their signal comigrated with the COA3 protein. The shift of COA3 migration in KO clones to the region around 130 kDa in COX4I1/4I2 KO likely represents the cIV assembly intermediate that was detected with MT-CO1 antibody in WB experiments (Figure 3a), as COA3 is a MT-CO1-interacting protein [20]. In accordance with BN-PAGE WB analysis and proteomic LFQ studies, SILAC-based complexome profiling confirmed the profound decrease in complex I displayed by the KO cells, being decreased approximately fivefold relative to control HEK293. In control cells, complex I was distributed between respirasome (I III_2_IV) and I III_2_ supercomplexes. Interestingly, the low amounts of fully assembled cI in the KO were found to either be associated to cIII in the I III_2_ supercomplex, or as cI assembly intermediates migrating at ≈750 kDa. These species contained subunits of all cI subunits corresponding to the different structural modules except for the N-module components, and it therefore corresponds to the cI precomplex, a stable intermediate formed before the addition of the most distal part of peripheral arm harboring the flavin mononucleotide site. While we did not detect co-migration of NDUFAF2, the typical marker of this Q/P subassembly [35], we did detect the accumulation of NDUFAF4 and ACAD9, which also associate with Q/P module and are released when cI assembly is completed [2]. Consistent with the results of WB and immunodetection and LFQ proteomics, only a modest decrease in complex III was observed in KO cells. Moreover, in agreement with BN-PAGE WB data, cIII distribution in COX4I1/4I2 KO shifted from supercomplexes to the cIII dimer. Level of cII was unchanged between control and KO cells. Distribution and content of cV corresponded to BN-PAGE WB experiments, and cV monomer was the dominant form detected both in control and COX4I1/4I2 KO samples. Minor accumulation of F_1_ subassemblies was observed in KO cells (Appendix A).

### 3.4. Decreased Rate of Mitochondrial Protein Synthesis in COX4I1-Lacking Cells

Pulse-chase in vivo ^35^S Met + ^35^S Cys metabolic labeling of mtDNA-encoded OXPHOS subunits was performed to decipher whether cIV and cI deficiencies detected in electrophoretic analyses were due to subunit degradation following assembly impairment, or rather caused by a defect in earlier steps of protein expression. Labeled proteins were separated by SDS-PAGE and detected by phosphor storage screen fluorescence (Figure 5a). These pulse-chase experiments revealed that apart from the MT-ATP6 and MT-ATP8 components of cV, the majority of mtDNA-encoded OXPHOS subunits were synthesized less in COX4I1 KO and COX4I1/4I2 KO clones than in the wt HEK293 cells in the “pulse” samples. Again, COX4I2 KO clones were not affected, having synthesis rates and protein patterns comparable to the wild-type. Detection of labeled proteins following the 24 h chase in non-labeling medium allowed for the estimation of protein turnover. In general, levels of labeled proteins were decreasing compared to pulse in all analyzed cell lines (Figure 5a). Strikingly, the MT-CO2/MT-CO3 double band completely disappeared during the chase period in COX4I1 and COX4I1/4I2 KO clones (Figure 5a, Appendix A), reflecting their degradation due to an inability to assemble, as suggested by previous data. In contrast, residual levels of MT-CO1 were still detectable after the 24 h chase. Concerning other OXPHOS complexes, levels of cIII and cV components displayed higher stability than cI (MT-ND1, MT-ND2) and cIV (MT-CO1-3) subunits in cells lacking COX4I1 (Figure 5a, Appendix A), in agreement with data on steady-state levels of these complexes. Therefore, cIV deficiency caused by COX4I1 gene KO may have deleterious effect on mitochondrial proteostasis at multiple levels, affecting the biogenesis of the OXPHOS complexes, especially cIV and cI, from synthesis to assembly.

Interestingly, the MS LFQ dataset provided further support for the hypothesis of disturbed mitochondrial translation, as content of mitochondrial ribosomal proteins showed a generalized decrease of about 1.5-fold in COX4I1 and COX4I1/4I2 KO clones (Figure 5b, Appendix A). Moreover, content of ribosomal assembly factors DDX28 and ERAL1 was decreased twofold in COX4I1-lacking cells (Figure 5b). MS LFQ data were confirmed in the SILAC/complexome profiling experiment, which also showed 1.5- to 2-fold reduction of both large and small mitoribosome subunits in COX4I1/4I2 KO (clone 2) in comparison with control HEK293 (Figure 5c).

## 4. Discussion

Previously, loss of function (LOF) models of OXPHOS genes in mammalian cells were relying on RNA interference (RNAi) technologies or deriving embryonic fibroblasts from mouse knockout animals (MEFs). The arrival of the relatively straightforward CRISPR-Cas9 gene editing technology [41] conveniently broadened experimental repertoire for construction of LOF models in human cell lines, avoiding the problems with residual content of the targeted protein encountered in the case of RNAi.

A high-throughput study employing CRISPR death screen searching for OXPHOS essential genes in human myelogenous leukemia K562 cells identified COX4I1 as KO-lethal under conditions when galactose was the sole carbohydrate source [50]. Using a targeted approach, we prepared single COX4I1 and COX4I2 KO cells, as well as double COX4I1/4I2 KO of both isoforms in HEK293 cells. Even under standard culture conditions, cultivation of COX4I1-lacking clones represented a significant challenge due to their highly glycolytic phenotype. The proliferation of KO cells was improved using richer DMEM/F-12 medium smented with uridine. Therefore, careful selection of a metabolically flexible cell line and optimization of culturing conditions is necessary for successful experimentation with knock-outs of essential OXPHOS genes.

As COX4 is an early incorporating subunit [4,13,16,51], impaired assembly of cIV enzyme was expected in COX4I1 and COX4I1/4I2 KO clones. Indeed, native electrophoresis confirmed presence of MT-CO1-containing cIV assembly intermediate, most likely representing a MITRAC complex [20]. Fully assembled cIV monomer and dimer were undetected. Moreover, the complete absence of fully assembled cIV was coupled with loss of supercomplexes containing cIV-III_2_IV and I III_2_IV (respirasome). Thus, our model cell lines unambiguously confirmed that COX4I1 subunit is essential for enzyme assembly process.

The possible compensation of COX4I1 isoform loss by expression of the second isoform, COX4I2, was tested, as isoform replacement mechanism was described while lost COX7A1 was substituted by its partner, COX7A2 isoform, in COX7A1 KO mice heart [52]. However, the COX4I2 isoform was not detected even after COX4I1 gene KO, and thus substitution of missing COX4 isoform 1 by COX4I2 isoform was not activated in HEK293 cells. Therefore, switch of COX4 isoforms seems to depend strictly on oxygen-regulated expression [53,54] and was not triggered by energy crisis in the studied cell line. A COX4 isoform switch in HEK293 cells is also improbable since COX4I2 isoform is not expressed in this cell line under normoxia. However, when the expression was forced by transfecting COX4I2 on the COX4I1 KO background, the second isoform was able to complement the cIV defect both at the structural and functional levels, as also seen in our previous study [15].

The catalytic function of cIV enzyme in COX4I1 KO clones was expected to be completely impaired due to loss of catalytic core mtDNA-encoded subunit MT-CO2, indispensable for cIV catalytic function [55]. The results of Seahorse measurement confirmed our hypothesis, as COX4I1 KO and COX4I1/4I2 KO clones demonstrated complete absence of oxygen consumption (OCR) and thus OXPHOS activity. Impaired OXPHOS function was compensated in COX4I1 KO and COX4I1/4I2 KO clones by increased glycolytic activity (ECAR). Therefore, ATP production is quite independent on OXPHOS and energy demands of COX4I1 KO cells are covered by ATP production on substrate level in glycolysis.

Since COX4I2 is not normally expressed in HEK293 cells cultured in normoxic conditions, the assembly processes of cIV and other OXPHOS complexes (cI, cII, cIII, cV) were not affected by COX4I2 gene KO. Moreover, cIV-containing supercomplexes were detected in COX4I2 KO clones comparably to wild-type. Furthermore, COX4I2 gene KO did not have any impact on mitochondrial respiration, as metabolic phenotype determined by Seahorse measurements was comparable to wild-type. Thus, no significant and functionally relevant changes were observed within COX4I2 KO clones in comparison with wild-type HEK293.

The most affected OXPHOS component apart from cIV in COX4I1 KO and COX4I1/4I2 KO clones was cI, as shown by NDUFA9 immunodetection in BN-PAGE WB and more comprehensively by both label-free and SILAC-based quantitative proteomics. Previous studies reporting similar cases of cIV–cI biogenesis interdependency in mouse cells caused by knock-out of COX10 [36], or by knock-down of COX4 expression [37], concluded that cIV absence caused defect in cI assembly. These findings were in line with the hypotheses that complexes I, III, and IV need to associate into supercomplexes to successfully proceed through their interdependent assembly pathways [34,35]. Recently, this hypothesis was confirmed by study showing that loss of cIII mtDNA-encoded subunit MT-CYB impairs assembly of not only cIII but also of cI and cIV. Specifically, cI assembly was stalled in late assembly intermediate before addition of the matrix-facing N-module, which occurs already in pre-respirasome association harboring cIII and also part of cIV [35]. The accumulation of cI assembly intermediates and the fact that residual mature cI was only found in association with III_2_ was also evidenced in our COX4I1/4I2 KO cell line using complexome profiling. This indicated that impaired cI assembly was involved in this case of cI-deficiency due to the lack of cIV assembly. In contrast to recent findings [40], in our BN-PAGE or complexome profiling experiments, we did not observe association of the residual cIV subunits or their assemblies with the I III_2_ supercomplex. In that study, ΔMT-CO2 cybrids harbored non-canonical subassembly of MT-CO1, COX5B, and HIGD2A binding to IIII_2_ and preventing its turnover [40]. Even though these subunits were not completely absent in COX4I1 lacking HEK293, their content was decreased in comparison with the ΔMT-CO2 cybrids, perhaps to a level unable to mediate the I III_2_ stabilization. It is also possible that COX4I1 absence prevents assembly of this non-canonical subcomplex. Nevertheless, impairment of cI assembly or stability might not be the only components of the observed cI deficiency, as our data further indicate that the interdependency started already at earlier levels of OXPHOS complex biogenesis. Indeed, the rate of mitochondrial protein synthesis was severely reduced both in COX4I1 single or COX4I1/4I2 knock-out cells. Notably, the synthesis of cIV and cI subunits was more affected than cIII MT-CYB and especially the mtDNA-encoded subunits of cV. This might reflect the recently reported spatial organization of mitochondrial translation [56], where translation of cV subunits occurs at separate mitoribosomes to facilitate their insertion directly into cristae membranes. Nevertheless, attenuation of mitochondrial translation was not observed in previously published cases of cIV–cI interdependency [35,36,37,38], indicating that complete absence of COX4 subunit affects cIV biogenesis at an earlier step than other cIV defects. This might be due to assembly-controlled translational plasticity realized by MITRAC complex, where absence of early assembly nuclear-encoded subunits such as COX4 leads to arrest of translation of MT-CO1 subunit that remains stalled as a ribosome-nascent chain complex containing MITRAC component COX14 [57]. Therefore, cIV assembly cannot proceed. Moreover, a portion of mitochondrial ribosomes might stay occupied by MT-CO1 transcripts, which could explain decreased synthesis of other cIV and cI subunits. In cases of interdependency on the basis of COX10 and MT-CO1 mutations, COX4 subunit was detectable and MT-CO1 synthesis was proceeding without translation stalling [36,38]. The cIV defect was instead associated with MT-CO1 destabilization due to defective hemylation or C-terminus truncation. Therefore, another possible link between early phases of cI and cIV biogenesis may be represented by COA1 protein. This mammalian homologue of yeast COX1 translational regulator COA1 was initially identified as a cIV biogenesis factor [20]. However, it was recently shown that it also serves as a translational factor of cI ND2 subunit in a similar manner to COX14 function during MT-CO1 synthesis [58]. COA1 may coordinate biogenesis of cIV and cI as a bifunctional protein to ensure production of corresponding amount of these respiratory complexes. One could speculate that stalling of MT-CO1 within MITRAC complex may negatively influence the availability of COA1 for MT-ND2 synthesis. The ND2 assembly module of complex I serves as a core for addition of other assembly modules [4], and therefore its deficiency would have a significant effect on cI content. Nevertheless, our data indicate that attenuation of cIV and cI mtDNA-encoded subunits is generalized. This could be explained by the decreased content of mitochondrial ribosomal proteins identified by MS LFQ analysis in COX4I1 and COX4I1/4I2 KO cells, as well as by complexome profiling in COX4I1/4I2 KO. The putative mitoribosome insufficiency could be associated with decreased content of ribosomal assembly factors DDX28 and ERAL1 detected in MS LFQ dataset. Contribution of other secondary mechanisms negatively influencing mitoribosome levels should also be considered. Energetic insufficiency or changes in redox status caused by mitochondrial defects can trigger integrated stress response controlled by the ATF4 transcription factor, which was shown to involve reduction of the content of mitoribosome protein components and subsequent downregulation of mitochondrial protein synthesis [59]. However, residual cIV amounts are sufficient to prevent an associated defect in cI, as indicated previously by study using knock-down of COX4 in HEK293 cells, which displayed unchanged levels of complex I [60]. Complete clarification of the observed interdependency warrants further studies.

In summary, CRISPR-Cas9-mediated knock-out of COX4 isoforms confirmed its position as the initiator subunit of cIV assembly. In normoxia, the COX4I2 isoform is not significantly expressed in HEK293, nor is it induced upon deletion of isoform COX4I1, and therefore COX4I1 single KO and COX4I1/4I2 KO cells displayed the same phenotype. Importantly, knock-in of either isoform 1 or 2 into COX4I1/4I2 KO was able to complement cIV defect, indicating that even COX4I2 is able to initiate de novo enzyme assembly. COX4I1-lacking cells presented with cIV–cI biogenesis interdependency associated with yet a novel mechanism involving the decrease in the amounts of mitochondrial ribosomes and the downregulation of mitochondrial protein synthesis. This illustrates the complicated and intertwined mechanisms governing the biogenesis of mitochondrial respiratory chain complexes and opens the ground for further elucidation of such mechanisms.

## Figures and Tables

**Figure 1 cells-10-00369-f001:**
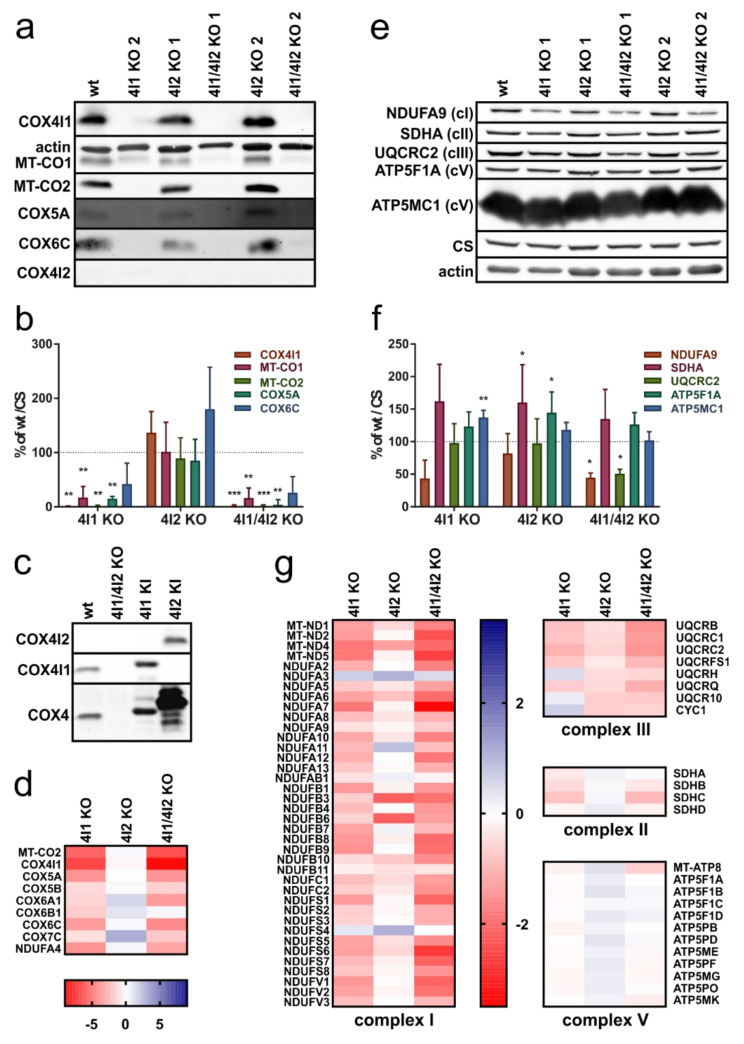
(**a**) Representative Western blot analysis of cytochrome c oxidase (cIV) subunits in KO clones. Whole cell lysates (30 µg of protein) of HEK293 (wild-type (wt)), COX4I1 KO, COX4I2 KO, and COX4I1/4I2 KO clones were subjected to SDS-PAGE and Western blotting. Actin and citrate synthase (CS) antibodies were used as a loading control. (**b**) Quantification of cIV subunit content normalized to citrate synthase. Data represent mean ± SD of wt (*n* = 2), COX4I1 KO (*n* = 4), COX4I2 KO (*n* = 4), and COX4I1/4I2 KO cells (*n* = 3). Statistically significant differences between wt and KO cells were calculated using ANOVA in GraphPad Prism 8. Asterisks (*) represent *p*-value: * < 0.05; ** < 0.01; *** < 0.001. (**c**) COX4I2 antibody testing. Whole cell lysates (30 µg of protein) of HEK293 (wt), COX4I1/4I2 KO, and COX4I1/4I2 KO, transfected with cDNA constructs of either COX4I1 (COX4I1 KI) or COX4I2 (COX4I2 KI), were subjected to SDS-PAGE and Western blotting followed by detection with antibodies specific for COX4 isoforms (COX4I1 and COX4I2, respectively), as well as with polyclonal antibody detecting both COX4 isoforms. (**d**) Proteomic analysis of relative content of cIV subunits using mass spectrometry label-free quantification (MS LFQ). Figure shows decreased content of cIV subunits in COX4I1 and COX4I1/COX4I2 KO clones, with MT-CO2 being the most affected. COX4I2 gene KO does not have deleterious effect on cIV subunit level. Data in heat map represent log_2_ values of quantity fold changes relative to wild-type, according to color scale shown below the heat map. (**e**) Western blot analysis of representative oxidative phosphorylation (OXPHOS) subunits for complexes I, II, III, and V. Whole cell lysates (30 µg of protein) of HEK293 (wt), COX4I1 KO, COX4I2 KO, and COX4I1/4I2 KO clones were subjected to SDS-PAGE. Immunodetected actin and citrate synthase were used as loading controls. The antibodies used are listed in Appendix A. (**f**) Quantification of OXPHOS complex subunits content normalized to citrate synthase. Data represent mean ± SD of wt (*n* = 2), COX4I1 KO (*n* = 4), COX4I2 KO (*n* = 4), and COX4I1/4I2 KO cells (*n* = 3). Statistically significant differences between wt and KO cells were calculated using ANOVA in GraphPad Prism 8. Asterisks (*) represent *p*-value: * < 0.05; ** < 0.01; *** < 0.001. (**g**) Proteomic analysis of OXPHOS complex subunits using mass spectrometry label-free quantification (MS LFQ). Data in heat map represent log_2_ values of quantity fold changes relative to wild-type, according to the color scale shown at the respective heat maps.

**Figure 2 cells-10-00369-f002:**
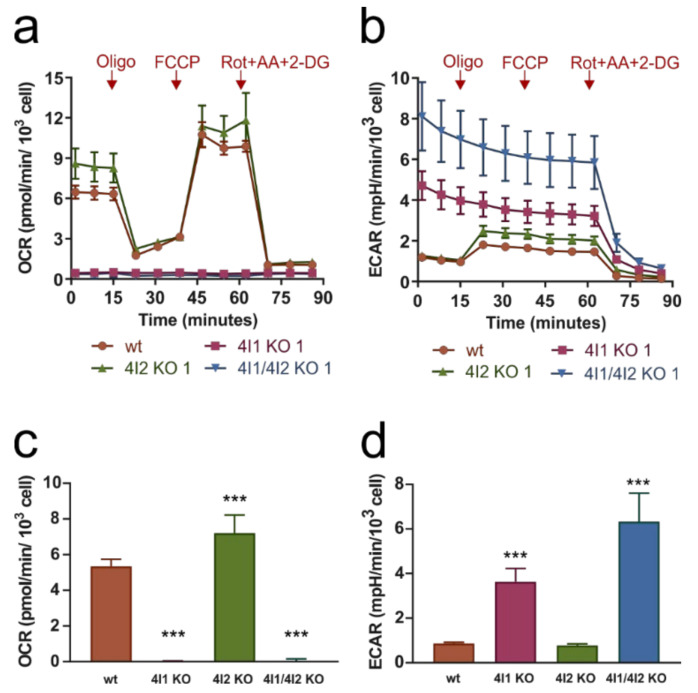
Parallel measurements of (**a**) cellular oxygen consumption rate (OCR) and (**b**) extracellular acidification rate (ECAR) were performed by Seahorse XF Bioenergetic Analyzer with 10 mM glucose as a substrate, and after subsequent additions of 1 µM oligomycin (Oligo); 1 µM FCCP; and inhibitor cocktail of 1 µM rotenone, 1 µg/mL antimycin A, and 100 mM 2-deoxyglucose (Rot + AA + 2DG). Data are plotted as mean values ± SD of independent experiments (biological replicates) for control HEK293 cells (wt, *n* = 5), COX4I1 KO clone 1 (*n* = 5), COX4I2 KO clone 1 (*n* = 4), and COX4I1/4I2 KO clone 1 (*n* = 3) and demonstrate complete absence of mitochondrial OXPHOS activity in COX4I1 and COX4I1/4I2 KO clones. (**c**) OCR and (**d**) ECAR bar graphs represent mean ± SD basal rates after subtraction of background rates in control HEK293 cells (wt, *n* = 5), COX4I1 KO clone 1 (*n* = 5), COX4I2 KO clone 1 (*n* = 4), and COX4I1/4I2 KO clone 1 (*n* = 3). Statistically significant difference of KO clones versus wild-type HEK293 were calculated using ANOVA in GraphPad Prism 8. Asterisks (*) represent *p*-value: *** < 0.001.

**Figure 3 cells-10-00369-f003:**
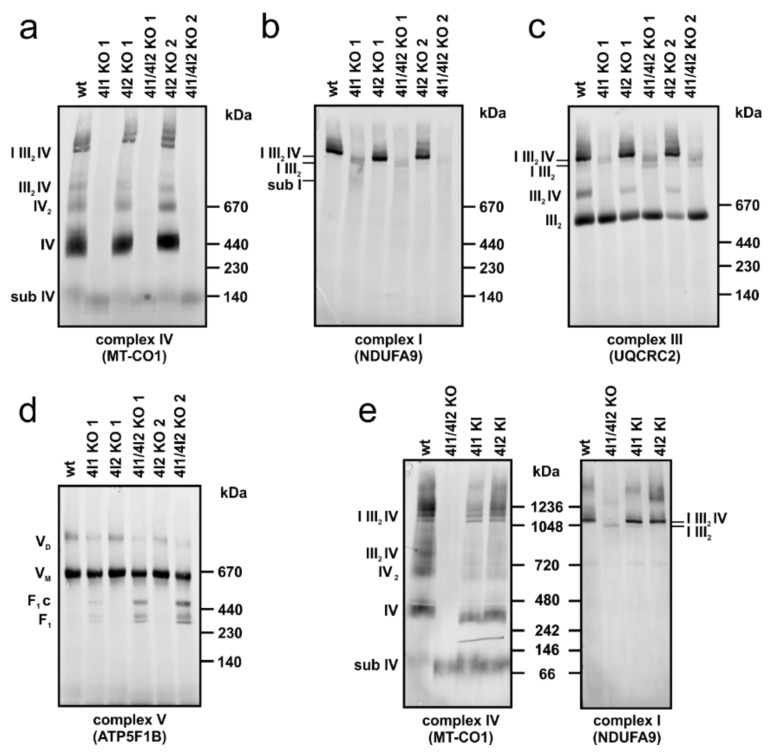
Western blot analysis of OXPHOS complexes and supercomplexes. Representative images of three independent analyses are shown. Digitonin solubilized mitochondria (30 µg of protein) of HEK293 (wt), COX4I1 KO, COX4I2 KO, and COX4I1/4I2 KO clones were subjected to blue-native (BN)-PAGE to detect OXPHOS complexes: (**a**) complex IV using MT-CO1 antibody, (**b**) complex I using NDUFA9 antibody, (**c**) complex III using UQCRC2 antibody, (**d**) complex V using ATP5F1B antibody. (**e**) Complex IV and complex I content and assembly status comparison of COX4I1/4I2 KO and COX4I1/4I2 KO cells rescued by overexpression of either COX4I1 (4I1 KI) or COX4I2 KI (4I2 KI). The antibodies used are listed in Appendix A.

**Figure 4 cells-10-00369-f004:**
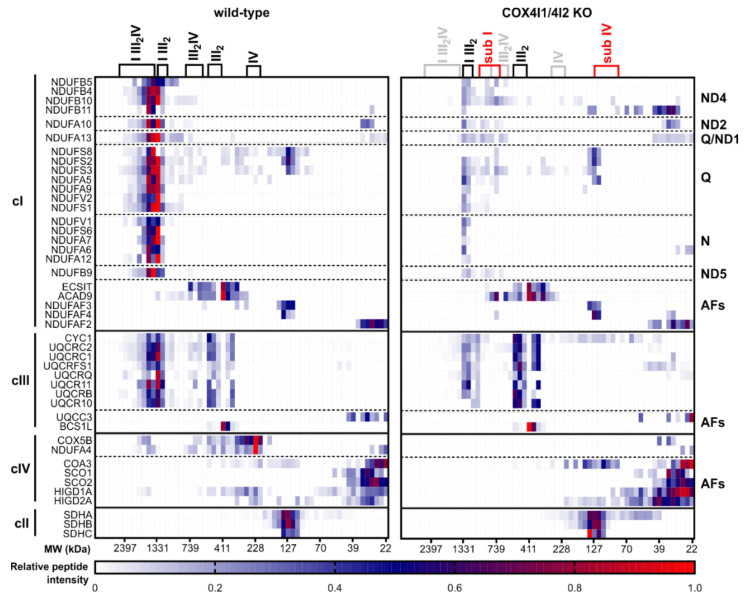
Complexome profiling analysis of OXPHOS complexes content and assembly status. Heatmap displays relative content and migration of respiratory chain complex components and their assembly factors under native conditions in control HEK293 and COX4I1/4I2 KO cell lines. The major entities of NADH dehydrogenase (cI), cytochrome *bc1* complex (cIII), and cIV detected in wt HEK293 are shown in the legend above the heat map. For COX4I1/4I2 KO, complexes retaining comparable content and migration as in the wt are denoted in the legend in black, the missing species are in grey, and the accumulated subcomplexes of cI (sub I) and cIV (sub IV) are marked red in the legend. Data represent relative peptide intensity profile of each protein normalized to the slice with the highest intensity, averaged from duplicate experiments, according to the color scale below the heatmap.

**Figure 5 cells-10-00369-f005:**
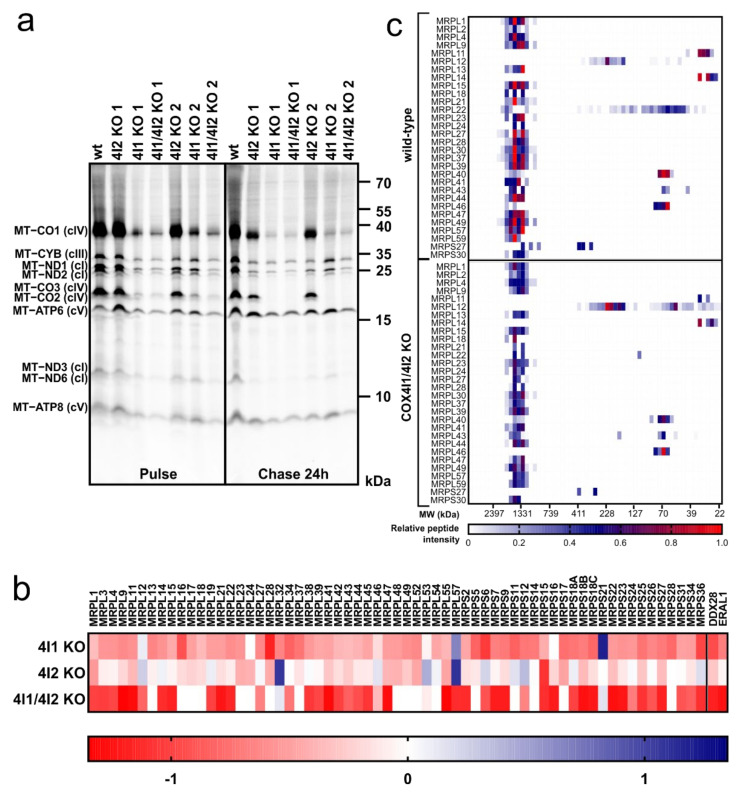
Mitochondrial protein synthesis. (**a**) ^35^S in vivo labeling of mtDNA-encoded OXPHOS subunits. Representative image of two independent experiments showing autoradiographic detection of labeled proteins in 60 µg protein of whole cell lysates analyzed by SDS-PAGE uncovers lower “pulse” level of most mtDNA-encoded proteins due to COX4I1 gene KO. (**b**) MS LFQ analysis revealed lower quantity of mitochondrial ribosomal proteins in COX4I1 and COX4I1/4I2 KO clones. COX4I2 KO clones were comparable to wild-type HEK293 cell line. Data in heat map represent log_2_ values of quantity fold changes relative to wild-type. (**c**) SILAC/complexome profiling of mitochondrial ribosomal proteins. Heatmap displays relative content and migration of proteins constituting large and small subunits of mitochondrial ribosome. Data represent relative peptide intensity profile of each protein normalized to the slice with the highest intensity, averaged from duplicate experiments.

## Data Availability

The data presented in this study are openly available in PRIDE repository, accession number PXD023367 (MS-LFQ dataset), and in CEDAR database, accession number CRX25 (complexome profiling data).

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
