# Peer review of "Loss of COX4I1 Leads to Combined Respiratory Chain Deficiency and Impaired Mitochondrial Protein Synthesis"

_cells, 2021, doi:10.3390/cells10020369_

Round 1

Reviewer 1 Report

This manuscript investigates how absence of 2 cytochrome c oxidase genes affects assembly of complex IV. in addition it investigates how a deficiency of complex IV affects mitochondrial respiration, the assembly of respirasomes, and the synthesis of mitochondrial proteins.

Overall, this manuscript offers some valuable ideas of the effect of a COX4i1 knock-out on the assembly of CxIV and mitochondrial supercomplexes.

Major: The paper is lengthy and descriptive. The introduction provides information that are not supportive or necessary for the aim of this paper.

Methods can be shorten by using references of the already published protocols; for example paragraphs 2.2, 2.3, and 2.4.

Figure 1: I would like to see the analysis shown in Figure S1 in the main manuscript. Figure S1: what type of analysis was used to establish significance with n=2? Please add the n for the experiments presented in Figure 1.

Some analysis is missing (Figure 2). Raw data are shown, but it is difficult to see, where statistically significant differences are. Can this figure be presented more concisely?  

Figure 2b COX4i1/4i2: n=3, but giant error bars, perhaps additional experiments could make these error bars smaller. Another possibility is to calculate the standard error (SE) instead of standard deviation (SD).

Figure 2: the text mentions Figure 2A, 2B and 2C, but the figure itself does not show which panel is A, B or C. Please add labeling.

How representative are the blots shown in Figure 3? Please add number of repeat experiments.

Figure 3: Panel B and the left image of panel E show a blot of NDUFA9. Please check the label for the supercomplexes in panel B, as it is not clear to me what is the monomer of CxI? It appears to me that this blot shows a problem with the gradient of the native gel, which increases the problem of interpreting this blot. Alternatively, why are the signals of the WT sample at a different molecular weight than the samples from 4i2 KO1 and 4i2KO2?

Line 78 and 391: While there is a discussion going on about NDUFA4 being a subunit of CxIV, there are also strong supportive data that NDUFA4 is a subunit of CxI and essential in the formation of respirasomes, and specifically the interaction of CxI and CxIV. The authors should add original research paper to support their hypothesis, although their own data support the conventional thought that NDUFA4 belongs to CxI.

Minor:

Consistency issues: for example the subfigures in Figure 1, 3, and 5 are labeled with A,B,C …., but the figure legends in 1, 2, 3 and 5 use a, b, c, … .

Methods:

Even though the method for obtaining HEK293 COX4i1 and COX4i2 knockouts were recently published (Reguera et al 2020), I would recommend a short summary of the method for easier understanding of the presented experiments.

What is used for these experiments? Isolated mitochondria from HEK293 cells? A cell homogenate/lysate? Please add protocols or reference protocol if already published.

Author Response

Response to reviewer 1

We thank the reviewer for valuable comments aiming to improve the clarity or results presentation and overall manuscript quality. Before addressing reviewer’s comments specifically, I would like to mention that based on the request of the reviewer #2, all the protein names throughout the manuscript were changed according HUGO gene nomenclature committee (https://www.genenames.org/), which also affects the marking of our cell line models.

Major: The paper is lengthy and descriptive. The introduction provides information that are not supportive or necessary for the aim of this paper.

We shortened the introduction. We removed the part dealing with the details of COX1 maturation within the MITRAC complex (lines 89-93 in the original manuscript), as well as the paragraph describing putative effect of changes in cellular redox status on cI biogenesis (lines 104-108), as these issues are not addressed by experiments of our study.

Furthermore, the part introducing the cases of interdependency of respiratory chain complexes interdependency (lines 63-75) was moved after the description of assembly of complexes IV and I, starting new paragraph at the line 110 in the original text.

Methods can be shorten by using references of the already published protocols; for example paragraphs 2.2, 2.3, and 2.4.

The paragraphs 2.2, 2.3, and 2.4 describing standard electrophoretic and Western blot procedures were significantly shortened using relevant references to published protocols.

Figure 1: I would like to see the analysis shown in Figure S1 in the main manuscript. Figure S1: what type of analysis was used to establish significance with n=2? Please add the n for the experiments presented in Figure 1.

We transferred part of the SDS/PAGE WB quantification (data normalized to citrate synthase) results into Figure 1 (c, d), the data normalized to actin were left as Supplementary figure 1. The data were originally statistically analyzed using t-test (calculated individually for each knock-out versus control). For the revision, we reanalyzed the data using more appropriate ANOVA, with wt n=2, COX4I1 KO n=4, COX4I2 KO n=4, and COX4I1/4I2 KO n=3. The Western blots shown in Figure 1 (a, b) are representative images of data used for the quantification.

Regarding the data from MS-LFQ shown in the heat maps in Figure 1, the research design is described in Methods part 2.5: Biological duplicates (COX4I1 KO clones 1 and 2, COX4I2 KO clones 1 and 2, COX4I1/4I2 KO clones 1 and 2) were harvested and MS LFQ analysis was performed in technical duplicates as described [42]. For the revision, the data are further shown in Supplementary Figure 2, expressed as average fold changes between individual KO cell lines and HEK293 wt for respiratory chain complexes, proteins of mitochondrial ribosomes, and all identified mitochondrial proteins (Mitocarta 2.0 annotated). Moreover, the data are available for review in the PRIDE repository, under the following access:

Username: reviewer_pxd023367@ebi.ac.uk

Password: EbAHZlbT

The data will be made public following eventual publication of the manuscript.

Some analysis is missing (Figure 2). Raw data are shown, but it is difficult to see, where statistically significant differences are. Can this figure be presented more concisely?  

We added bar graphs showing mean ± SD for OCR and ECAR under basal condition (before the addition of oligomycin). ANOVA was used for statistical analysis. Compared to original submission and based on suggestion by reviewer #2, we omitted the panel c of Figure 2, showing OCR complementation by COX4I1 or COX4I2 knock-in on the COX4I1/4I2 KO background, as comparable results obtained by independent measurements were already published in our previous study (Reguera et al. 2020).

Figure 2b COX4i1/4i2: n=3, but giant error bars, perhaps additional experiments could make these error bars smaller. Another possibility is to calculate the standard error (SE) instead of standard deviation (SD).

The huge error bars of ECAR data of COX4I/4I2 KO were partially normalized by subtraction of the background signal (after rotenon + antimycin A + 2-deoxyglucose addition) for presentation as bar graph in 2d. But perhaps there may be rational explanation for huge variation of this parameter – due to total deficiency of cIV, the metabolic pathways have to completely readapted in COX4 KO cell lines to allow for sufficient ATP production by substrate phosphorylations and reoxidation of NADH. This is accompanied by increased production and release of lactate, and perhaps of other acidic intermediates into the medium. Our undestanding of the metabolic rearrangements is still scarce, but it is possible that excretion of the metabolic waste products can be diverse leading to variation in extracellular acidification rate.

Figure 2: the text mentions Figure 2A, 2B and 2C, but the figure itself does not show which panel is A, B or C. Please add labeling.

We apologize for the missing labelling in the original submission, we added it for the revised version.

How representative are the blots shown in Figure 3? Please add number of repeat experiments.

The images are representatives of three independent experiments, this information was added into Figure 3 legend.

Figure 3: Panel B and the left image of panel E show a blot of NDUFA9. Please check the label for the supercomplexes in panel B, as it is not clear to me what is the monomer of CxI? It appears to me that this blot shows a problem with the gradient of the native gel, which increases the problem of interpreting this blot. Alternatively, why are the signals of the WT sample at a different molecular weight than the samples from 4i2 KO1 and 4i2KO2?

We are grateful for this crucial comment, we indeed mislabeled the cI complexes in Figure 3b, and we admit that the upper part of gradient of BN gels are bit uneven. Importantly, in HEK293 wt as well as in COX4I2 KO, cI is exclusively present in respiratory supercomplexes (labelled as I III2 IV). In cell lines lacking COX4I1, cI is mostly found in association with cIII (I III2) and there is weak band representing an assembly subcomplex of cI (sub cI). Free complex I was in fact not detected in these samples. The pattern of cI complexes can be also appreciated in Figure 4 showing the complexome profiling results, the new labelling of Figure 3b is now consistent with the pattern shown in Figure 4.

Also in this regard, we replaced the left image in Fig. 3e (detection with MT-CO1 antibody) for a one with more comparable gradient to the NDUFA9 image, so that an easier assignment of the respiratory supercomplex I III2 IV can be made.

Line 78 and 391: While there is a discussion going on about NDUFA4 being a subunit of CxIV, there are also strong supportive data that NDUFA4 is a subunit of CxI and essential in the formation of respirasomes, and specifically the interaction of CxI and CxIV. The authors should add original research paper to support their hypothesis, although their own data support the conventional thought that NDUFA4 belongs to CxI.

We agree, that there is still ongoing discussion regarding assignment of NDUFA4 into cI or cIV. Therefore, we added new references into the introduction where cIV composition is described (line 79 of the original text), that NDUFA4 is considered as part of cIV based on genetic experiments (Balsa et al 2012) or cryo-EM structural data (Zong et al. 2018). In fact, our data are more supportive for NDUFA4 being part of cIV, especially the complexome profiling data (Figure 4) is quite revealing in this regard – the NDUFA4 has very comparable pattern to COX5B in wt sample. In COX4I1/4I2 sample, the NDUFA4 is detected only as a low molecular entity and is not associated with I III2 supercomplex or in the cI subcomplex (sub I) as other cI subunits. To avoid confusion, the protein reported as associating with cI subcomplex (line 406 in the original text) is NDUFAF4, the cI assembly factor.

Minor:

Consistency issues: for example the subfigures in Figure 1, 3, and 5 are labeled with A,B,C …., but the figure legends in 1, 2, 3 and 5 use a, b, c, … .

The subfigures are now consistently labeled with lowercase letters.

Even though the method for obtaining HEK293 COX4i1 and COX4i2 knockouts were recently published (Reguera et al 2020), I would recommend a short summary of the method for easier understanding of the presented experiments.

The following text was added to Methods part 2.1:

Briefly, COX4I1 and COX4I2 knock-outs were introduced by CRISPR knock-out technology employing Cas9-D10A paired nickase with two chimeric RNA duplexes. To obtain double knock-out of both isoforms (COX4I1/4I2 KO), COX4I2 and COX4I1 were knocked-out subsequently. For this study, two individual clones of each knock-out was used (COX4I1 KO 1 and 2, COX4I2 KO 1 and 2, COX4I1/4I2 KO 1 and 2). For rescue experiments, COX4I1/4I2 KO 2 served as a background generating cell lines with stable expression of COX4I1 (COX4I1 KI) or COX4I2 (COX4I2 KI), using full length cDNA constructs in pcDNA3.1+ with C-terminal FLAG (DYKDDDDK) tag.

What is used for these experiments? Isolated mitochondria from HEK293 cells? A cell homogenate/lysate? Please add protocols or reference protocol if already published.

The sample type for each type of experiment was already described in the original submission at the relevant part of the Methods. Briefly, whole cell lysates were used for SDS-PAGE WB, MS-LFQ and SDS-PAGE analysis following in-vivo labelling with 35S methionine and cysteine. Isolated mitochondria were used for digitonin solubilizations for BN-PAGE and complexome profiling. In Seahorse experiments, OCR and ECAR rates were recorded from intact cells.

Reviewer 2 Report

The authors used HEK293 cells knocked out for the alternative COX4 subunits COX4i1, COX4i2 or both to study effects of cIV deficiency on respiration, respiratory chain complex and supercomplex assembly, and stability. A series of methods and approaches were used indicating that knock-out of the COX4I1 gene (COX4I2 is apparently not expressed in the HEK293 cells) impacts levels of cIV and cI proteins, cI and cIV assembly and assembly of supercomplexes containing these complexes. The manuscript has shortcomings in clarity of presentation, conciseness and there are a number of concerns regarding interpretation of the results. In the following specific points.

Nomenclature: Confusing synonymous use of terms: COX, cIV, unexplained COX1 deficiency of single gene… And not COX for cIV. In other communities COX stands for cyclooxygenase. In the beginning the different levels should be named as this is done in lines 46 to 49, but from then on it would be preferential to use cI, cII, cIII, cIV, and cV for the respiratory chain complexes throughout the text. As for the complex subunits: Less ambiguous, COX IV-1 is the recommended short protein name for what is called COX4i1 in the manuscript. Figure 1 C shows MT-CO2 and the text refers to COX2. To improve understandability for readers a consistent and non-ambiguous nomenclature would be helpful. Gene names are curated by the Human Gene Name Committee and one possibility for non-ambiguous acronyms is to use the gene HGNC gene names in non-italic capitals for the respective proteins. It is also helpful for identifying the mtDNA encoded subunits. At least there has to be made a clear explanation which nomenclature conventions are used and then they must be adhered to throughout the manuscript.

Also, the gene-modified HEK293 derivatives: it is not always clear from the labels in which background (wt, COX4i1 KO, COX4i2 KO, COX4i1/COX4i2 KO the respective KI’s are present: Better to write that out and make it very clear in the legend. For the labels in 1B, for example, it could be considered to put 4i1/4i2 KO horizontally over the 4i1 KI and 4i2 KI lane labels. Similarly, graph C in figure 2.

Introduction

The major question – how deficiency in either of the COX4 subunit isoforms impacts respiratory chain complexes and supercomplexes – the literature knowledge should be summed up more concisely. In the paragraph on cIV, the relationships and knowledge on COX4i1 and COX4i2 should be presented. It is not common knowledge that there are a series of alternative cIV subunits that are expressed in certain tissues. Some background especially on the alternative COX4i1 and COX4i2 subunits should be presented in the introduction.

Line 74 ‘….enzyme assembly..’ How does this distinguish from complex assembly?

Materials and methods

Mitochondrial protein synthesis analysis: number of replicates and repetitions (biological replicates) should be stated and the relationship of the pulse /chase pairs (see below).

Seahorse analyses: Number of well replicates and biological replicates (analysis on different days) are not clearly explained.

Results

Line 238 (results) ‘…we produced…’ as stated in mat met this was described in a previous publication and the clones were used here. The phrase suggests that these are novel clones produced.

Second paragraph of results: it is already evident from the previous publication (ref 38), COX4I2 is not detectable/very lowly expressed in wt HEK293 cells. This should be mentioned.

Line 247 ff; ‘…COX4i2 KO have mildly decreased number of mitochondria..’ Doubtful statement; only one subunit ‘significant’. Documentation of average fold change of MitoCarta proteins in the KOs would be informative (see also below).

Non-specific COX4 antibody: this is a supposedly anti-COX4i2 antibody raised against an antigen containing COX4i2 protein sequence. Does it cross-react with COX4i1? There is considerable sequence similarity between COX4i1 and COX4i2; this should be mentioned and commented. Figure 2 C shows that COX4i2 can functionally replace COX4i1. COX4i2 protein not detectable by WB in wt HEK293: Fig 1 B). Wrapping up that COX4i2 is not a relevant protein in HEK293 cells under the used conditions, but that it can functionally replace COX4i1 at some point of the manuscript would be clarifying.

Line 282: MS should be added: ‘…LFQ (MS) analysis..’

Lines 301-302: Legend to figure 1: better title for D): Western blot analysis of representative OXPHOS subunits for complexes I, II, II, and V.

Line 277-278: ‘…that in spite of the COX impairment, proteins can be efficiently imported into mitochondria.’ This is surprising: how can proton pumping to maintain the membrane potential, which is essential for protein import, be maintained in the absence of final electron transfer to oxygen? There lacks documentation for this statement, e.g. an average fold change for MitoCarta proteins in the KO cells compared to wt.

Figure 1: Capitals (A, B etc.) in Figure but a), b) etc. in legend. Choose one for both.

E): It appears that COX4i2 KO shows an opposite profile of cIII subunit changes compared to COX4i2. In the range of a log(2) fold changes of 1 that are taken as relevant for cI and cIV. Is this due to large variation in the quantitative data or could it mean something? Again: COX4i2 appears to be not or very lowly expressed in HEK293 cells and it is also not induced by COX4i1 KO, suggesting that HEK2903 cells only use COX4i1.

D): For helping the reader, the complexes that the subunits represent should be indicated in the figure, or at least in the legend.

3.2 Seahorse metabolic analysis: this is to a large part a repeat of what was published in reference 38.

Figure 2: Indexing: a, b, and c not indicated for the graphs. What are the n values standing for: number of wells or data from independent experiments (with ?number of wells)? The labels in 2C for the KI clones are ambiguous: one could understand that it is COX4i1 or COX4i2 KI on wt background.

3.3 Effects on complexes: this part provides strong evidence for an effect of COX4i1 KO, but not COX4i2 KO, on cI and supercomplexes containing cI and cIII. It also shows that both COX4 variants can functionally complement COX4i1 KO.

Author Response

Response to reviewer 2

We thank the reviewer for valuable comments aiming to improve the clarity or results presentation and overall manuscript quality.

 Nomenclature: Confusing synonymous use of terms: COX, cIV, unexplained COX1 deficiency of single gene… And not COX for cIV. In other communities COX stands for cyclooxygenase. In the beginning the different levels should be named as this is done in lines 46 to 49, but from then on it would be preferential to use cI, cII, cIII, cIV, and cV for the respiratory chain complexes throughout the text. As for the complex subunits: Less ambiguous, COX IV-1 is the recommended short protein name for what is called COX4i1 in the manuscript. Figure 1 C shows MT-CO2 and the text refers to COX2. To improve understandability for readers a consistent and non-ambiguous nomenclature would be helpful. Gene names are curated by the Human Gene Name Committee and one possibility for non-ambiguous acronyms is to use the gene HGNC gene names in non-italic capitals for the respective proteins. It is also helpful for identifying the mtDNA encoded subunits. At least there has to be made a clear explanation which nomenclature conventions are used and then they must be adhered to throughout the manuscript.

We followed the recommendation, and so throughout the manuscript we strictly denote cytochrome c oxidase as cIV, and accordingly all other OXPHOS complexes. To identify proteins, we changed all protein names in the manuscript according to the HGNC, including labels in all figures.

Also, the gene-modified HEK293 derivatives: it is not always clear from the labels in which background (wt, COX4i1 KO, COX4i2 KO, COX4i1/COX4i2 KO the respective KI’s are present: Better to write that out and make it very clear in the legend. For the labels in 1B, for example, it could be considered to put 4i1/4i2 KO horizontally over the 4i1 KI and 4i2 KI lane labels. Similarly, graph C in figure 2.

The COX4I1 and COX4I2 KI were generated only on COX4I1/4I2 KO background. To unambigously denote the cell models, we added the following paragraph into Methods, part 2.1:

“Briefly, COX4I1 and COX4I2 knock-outs were introduced by CRISPR knock-out technology employing Cas9-D10A paired nickase with two chimeric RNA duplexes. To obtain double knock-out of both isoforms (COX4I1/4I2 KO), COX4I2 and COX4I1 were knocked-out subsequently. For this study, two individual clones of each knock-out were used (COX4I1 KO 1 and 2, COX4I2 KO 1 and 2, COX4I1/4I2 KO 1 and 2). For rescue experiments, COX4I1/4I2 KO 2 served as a background generating cell lines with stable expression of COX4I1 (COX4I1 KI) or COX4I2 (COX4I2 KI), using full length cDNA constructs in pcDNA3.1+ with C-terminal FLAG (DYKDDDDK) tag.”

Also, the background for knock-ins is also specifically mentioned in the legend to Figure 3e, graph 2c was taken away in the revised manuscript.

 Introduction

The major question – how deficiency in either of the COX4 subunit isoforms impacts respiratory chain complexes and supercomplexes – the literature knowledge should be summed up more concisely. In the paragraph on cIV, the relationships and knowledge on COX4i1 and COX4i2 should be presented. It is not common knowledge that there are a series of alternative cIV subunits that are expressed in certain tissues. Some background especially on the alternative COX4i1 and COX4i2 subunits should be presented in the introduction.

We rearranged the introduction so that the part describing how deficiencies of single respiratory chain complex can affect the levels of other complexes, which was split in two parts (lines 63-75 and 108-115) in the original text, is now undivided and follows the description of cI assembly.

Regarding the COX4 isoforms, the following text was added (line 79 in the original text):

“In case of COX4 subunit specifically, alternative isoforms COX4I1 and COX4I2 are expressed by independent genes. COX4I1 represents the ubiquitous isoform, while COX4I2 is mainly lung-specific and its expression is further regulated by oxygen levels [23]. It was recently shown that COX4I2 is necessary for hypoxic pulmonary vasoconstriction [24], and that COX4 isoform exchange may modulate cIV affinity to oxygen [25].“

To make the manuscript more concise, we removed the part dealing with the details of COX1 maturation within the MITRAC complex (lines 89-93 in the original manuscript), as well as the paragraph describing putative effect of changes in cellular redox status on cI biogenesis (lines 104-108), as these issues are not addressed by experiments of our study.

Line 74 ‘….enzyme assembly..’ How does this distinguish from complex assembly?

It was used as a synonymous term, to avoid ambiguity, we changed “enzyme“ to “complex“.

Materials and methods

Mitochondrial protein synthesis analysis: number of replicates and repetitions (biological replicates) should be stated and the relationship of the pulse /chase pairs (see below).

We added the following sentence: “For this experiment, wild-type HEK293 two independent clones of COX4I1 KO, COX4I2 and COX4I1/4I2 KO were analyzed in two independent experiments.“

Regarding the pulse/chase pairs, each cell line was labelled in two parallel dishes, one for pulse and the other for chase sample. This was confusingly stated in the original text, so we modified the sentence to: “Parallel dish of labelled cells (“chase” samples) were further incubated in complete non-labelling DMEM medium for 24 hours.“

Seahorse analyses: Number of well replicates and biological replicates (analysis on different days) are not clearly explained.

We added the information about biological replicates: „We analyzed wild-type HEK293 (n=5 - biological replicates – measurement on different days), COX4I1 clone 1 (n=5), COX4I2 KO clone 1, and COX4I1/COX4I2 clone 1 (n=3).“ This information is also part of the Figure 2 legend.

Each biological replicate was analysed in 5 well (technical) replicates. The information was added as: “Briefly, one day prior to measurement, 3.104 cells were seeded in pentaplicates in poly-L-lysine coated wells of measuring plate, and incubated overnight under standard cultivation conditions.“

Results

Line 238 (results) ‘…we produced…’ as stated in mat met this was described in a previous publication and the clones were used here. The phrase suggests that these are novel clones produced.

This statement is indeed ambiguous. Although some cell lines were already used for our previous publication – COX4I2 KO and COX4I1 KO on the COX4I2 KO background (COX4I1/4I2 KO), the COX4I1 KO on wt background is used here for the first time. Nevertheless we used less definite expression and changed “we produced“ to “we utilized“.

Second paragraph of results: it is already evident from the previous publication (ref 38), COX4I2 is not detectable/very lowly expressed in wt HEK293 cells. This should be mentioned.

We modified the sentence: “As we showed previously [25], HEK293 cells (wt) do not show detectable levels COX4i24I2 protein.“

Line 247 ff; ‘…COX4i2 KO have mildly decreased number of mitochondria..’ Doubtful statement; only one subunit ‘significant’. Documentation of average fold change of MitoCarta proteins in the KOs would be informative (see also below).

We appreciate this crucial comment, the statement really is not fully supported by the data, so we omitted the sentence. We reanalyzed our MS-LFQ dataset and calculated the average fold change for all Mitocarta proteins, OXPHOS proteins, protein subunits of mitochondrial ribosome, and for each individual OXPXHOS complex – these data are shown in Supplement figure 2. The average fold changes of wt HEK293 and COX4I2 KO are indeed comparable.

Non-specific COX4 antibody: this is a supposedly anti-COX4i2 antibody raised against an antigen containing COX4i2 protein sequence. Does it cross-react with COX4i1? There is considerable sequence similarity between COX4i1 and COX4i2; this should be mentioned and commented. Figure 2 C shows that COX4i2 can functionally replace COX4i1. COX4i2 protein not detectable by WB in wt HEK293: Fig 1 B). Wrapping up that COX4i2 is not a relevant protein in HEK293 cells under the used conditions, but that it can functionally replace COX4i1 at some point of the manuscript would be clarifying.

The details of the antibody are given in Supplementary table 1. While this particular antibody is marketed as a anti-COX4I2 antibody, it crossreacts with COX4I1 as well. We denoted this antibody as „dual COX4 isoform-specific, and commented the dual specifity in Supplementary table 1.

The summarizing statement about the relevance of COX4I2 in HEK293 cells is part of the discussion (lines 484-493 in the original text)

Line 282: MS should be added: ‘…LFQ (MS) analysis..’

We changed accordingly.

Lines 301-302: Legend to figure 1: better title for D): Western blot analysis of representative OXPHOS subunits for complexes I, II, II, and V.

We modified the title accordingly. Please note, that due to the other changes, this is no longer Figure 1D, but now Figure 1b.

Line 277-278: ‘…that in spite of the COX impairment, proteins can be efficiently imported into mitochondria.’ This is surprising: how can proton pumping to maintain the membrane potential, which is essential for protein import, be maintained in the absence of final electron transfer to oxygen? There lacks documentation for this statement, e.g. an average fold change for MitoCarta proteins in the KO cells compared to wt.

We have unpublished preliminary data, that mitochondrial membrane potential is decreased in COX4 knockouts, but not completely diminished. Its maintanance may be achieved e.g. by reverse action of ATP synthase and adenine nucleotide translocator, accompanied by complex metabolic rearrangements necessary for redox cofactors regeneration and efficient excretion of waste metabolites.

As mentioned above, we calculated the average fold changes from the MS-LFQ dataset showing that Mitocarta-annotated proteins levels are decreased only slightly compared to severely affected levels of cIV and cI subunits.

The whole paragraph was modified taking into account the newly shown average fold changes:

„The Western blot-based quantification data were complemented by MS LFQ analysis of the overall cellular proteome Overall, we observed only slightly decreased quantity of Mitocarta-annotated proteins in COX4I1/4I2 KO (Fig. S2a, b) and COX4I1 KO (Fig. S2b), which indicates, that in spite of the cIV impairment, proteins can still be efficiently imported into mitochondria. However, it is also obvious, that levels of cI subunits show profound and uniform decrease in COX4I1/4I2 and COX4I1 KO (Fig. S2c). Full data for individual OXPHOS complexes are presented as heatmaps (Figure 1g), quantitating 38 cI subunits, all 4 subunits of cII, 8 subunits of cIII, and 12 subunits of cV. From the perspective of multi-subunit comparison, the decrease of NDUFA9 shown by Western blot analysis was confirmed by MS LFQ analysis. The levels of cI subunits were between 1.5 – 3-fold decreased in COX4I1 and COX4I1/4I2 KO clones, while levels of cI subunits in COX4I2 KO were comparable to wild-type. On average, levels of cIII subunits were slightly decreased in COX4I1/4i2 KO cells, but it was less apparent in COX4I1 KO (Fig. 1g, S2c). The remaining OXPHOS complexes cII, and cV were not altered in the COX4I1 and COX4I1/4I2 KO clones. Consistently, the levels of mitochondrial proteins overall as well as of individual OXPHOS complexes COX4I2 KO clones were not apparently affected, with the exception of cIII that was slightly decreased (Fig. 1g, S2b,c).”

Figure 1: Capitals (A, B etc.) in Figure but a), b) etc. in legend. Choose one for both.

Lowercase letters are now consistently used for figures.

E): It appears that COX4i2 KO shows an opposite profile of cIII subunit changes compared to COX4i2. In the range of a log(2) fold changes of 1 that are taken as relevant for cI and cIV. Is this due to large variation in the quantitative data or could it mean something? Again: COX4i2 appears to be not or very lowly expressed in HEK293 cells and it is also not induced by COX4i1 KO, suggesting that HEK2903 cells only use COX4i1.

When calculated as average fold change, levels of cIII subunits are slightly decreased in COX4i2 KO compared to wild-type, while all other groups show comparable values. This is difficult to explain, but it may represent stochastic variation fixed during production of single cell knock-out clones. In this regard, the differences between COX4I1 KO and COX4I1/4I2 KO may be due to similar phenomena.

Moreover, the MS-LFQ data are available for review in the PRIDE repository, under the following login:

Username: reviewer_pxd023367@ebi.ac.uk

Password: EbAHZlbT

The data will be made public following eventual publication of the manuscript.

D): For helping the reader, the complexes that the subunits represent should be indicated in the figure, or at least in the legend.

We added the complexes in the figure, and also later in Figure 5 showing the pulse-chase labelling pattern of mtDNA-encoded proteins.

3.2 Seahorse metabolic analysis: this is to a large part a repeat of what was published in reference 38.

The data presented here are from independent experiments than data from reference 38. Nevertheless, since the outcome of measurements with COX4I1 KI and COX4KI2 is similar, we omitted the original Fig. 2c in the revised version and refer to our previous publication only. Instead we added bar graphs showing OCR (Fig. 2c) and ECAR (Fig. 2d) in basal state more suitable for statistical comparison between wt, COX4I1 KO, COX4I2 KO and COX4I1/4I2 KO.

Figure 2: Indexing: a, b, and c not indicated for the graphs. What are the n values standing for: number of wells or data from independent experiments (with ?number of wells)? The labels in 2C for the KI clones are ambiguous: one could understand that it is COX4i1 or COX4i2 KI on wt background.

Indexing was fixed. The n values stand for biological replicates (independent experiments), this is now mentioned in methods and also explicitly stated in the figure legend. 

3.3 Effects on complexes: this part provides strong evidence for an effect of COX4i1 KO, but not COX4i2 KO, on cI and supercomplexes containing cI and cIII. It also shows that both COX4 variants can functionally complement COX4i1 KO.

We fully agree with this interpretation. We believe our text already expresses these exact ideas.

Round 2

Reviewer 2 Report

Abstract, last sentence:

‘We propose that impairment of mitochondrial protein synthesis caused by cIV deficiency represents one of the mechanisms, which may couple biogenesis of cI and cIV.’

The statement is a bit fuzzy. Maybe ‘jeopardize’ is more correct than ‘couple’. cIV deficiency, e.g. in the case of lack of expression of COX4 subunits, evidently affects cIV assembly. The novel finding is that cI assembly is affected in this condition, which may be explained with impaired mitochondrial protein synthesis.

Results:

‘…subunits were assessed in an unbiased and high-throughput manner using mass spectrometry …’

All methods have their limitations and some bias. MS has other limits than WB; one might thus say that respiratory chain subunit levels were assessed using an alternative method that is not limited to representative single subunits only.

Figure 4:

Why are there no data on cV?

Figure 6:

Should not generally decreased levels of mitochondrial ribosome subunits result in decreased synthesis of all mtDNA encoded genes in COX4i1 KO cells?

Discussion

Large parts of the discussion concerned with complex and supercomplex assembly are quite speculative. It would be interesting to also speculate about potential mechanisms that might cause the observed decreased expression of mitochondrial ribosomal subunits.

Lines464-465: ’.. while generation of MEFs is challenging in terms of resources.’ Why?

Author Response

RESPONSE TO REVIEWER #2, ROUND 2.

We thank the reviewer for valuable comments aiming to improve the clarity or results presentation and overall manuscript quality.

Abstract, last sentence:

‘We propose that impairment of mitochondrial protein synthesis caused by cIV deficiency represents one of the mechanisms, which may couple biogenesis of cI and cIV.’

The statement is a bit fuzzy. Maybe ‘jeopardize’ is more correct than ‘couple’. cIV deficiency, e.g. in the case of lack of expression of COX4 subunits, evidently affects cIV assembly. The novel finding is that cI assembly is affected in this condition, which may be explained with impaired mitochondrial protein synthesis.

          We rewrote the final sentence in the abstract:

„We propose that attenuation of mitochondrial protein synthesis caused by cIV deficiency represents one of the mechanisms, which may impair biogenesis of cI.“

Results:

‘…subunits were assessed in an unbiased and high-throughput manner using mass spectrometry …’

All methods have their limitations and some bias. MS has other limits than WB; one might thus say that respiratory chain subunit levels were assessed using an alternative method that is not limited to representative single subunits only.

We agree with comment on methods bias, we left out the „unbiased“ from the sentence:

„In addition to classical electrophoretic and immunodetection approach, relative levels of cIV subunits were assessed in a high-throughput manner using mass spectrometry label-free quantification (MS LFQ).“

Figure 4:

Why are there no data on cV?

We wanted to focus this figure on respiratory chain complexes only, also taking into account the legend above the heat map. Addition of the description of cV assemblies into the legend could have made it complicated due to migration overlap with the assemblies of complexes I, III, and IV. The data, however, are available and they correspond to results from BN-PAGE WB experiments. We now present the complexome profiling heat map for the detected cV subunits as supplementary figure 4. The results are briefly described and the figure is referenced as follows:

  „Distribution and content of cV corresponded to BN-PAGE WB experiments, cV monomer was the dominant form detected both in control and COX4I1/4I2 KO samples. Minor accumulation of F1 subassemblies was observed in KO cells (Supplementary figure 4).“

Figure 6:

Should not generally decreased levels of mitochondrial ribosome subunits result in decreased synthesis of all mtDNA encoded genes in COX4i1 KO cells?

Recent studies indicate that mitochondrial protein synthesis may not occur on a general pool of mitoribosomes, but rather separate pools of ribosomes might be spatially organized at different regions of inner mitochondrial membrane to facilitate co-translational insertion of cV subunits into cristae membranes, while components of respiratory chain subunits are translated and assembled at the inner boundary membrane. We included this in the discussion including the new reference 56:

„This might reflect the recently reported spatial organization of mitochondrial translation [56], where translation of cV subunits occurs at separate mitoribosomes to facilitate their insertion directly into cristae membranes.“

Discussion

Large parts of the discussion concerned with complex and supercomplex assembly are quite speculative. It would be interesting to also speculate about potential mechanisms that might cause the observed decreased expression of mitochondrial ribosomal subunits.

In the discussion, we included new reference to a study describing an integrated stress response under the control of transcription factor ATF4 that was previously shown to downregulate mitochondrial protein synthesis:

„Contribution of other secondary mechanisms negatively influencing mitoribosome levels should also be considered. Energetic insufficiency or changes in redox status caused by mitochondrial defects can trigger integrated stress response controlled by the ATF4 transcription factor, which was shown to involve reduction of the content of mitoribosome protein components and subsequent downregulation of mitochondrial protein synthesis [59].“

Lines464-465: ’.. while generation of MEFs is challenging in terms of resources.’ Why?

This was meant so that it is necessary to have the animal model first to generate MEFs subsequently. But we agree that it is unnecessary statement, so we rewrote the first paragraph of the discussion as follows:

“Previously, loss of function (LOF) models of OXPHOS genes in mammalian cells were relying on RNA interference (RNAi) technologies or deriving embryonic fibroblasts from mouse knockout animals (MEFs). The arrival of the relatively straightforward CRISPR-Cas9 gene editing technology [41] conveniently broadened experimental repertoire for construction of LOF models in human cell lines, avoiding the problems with residual content of the targeted protein encountered in case of RNAi.”
